# How Interdisciplinary Interventions Can Improve the Educational Process of Children Regarding the Nutritional Labeling of Foods

**DOI:** 10.3390/foods12234290

**Published:** 2023-11-28

**Authors:** Juliana de Lara Castagnoli, Elisvânia Freitas dos Santos, Daiana Novello

**Affiliations:** 1Department of Nutrition, Health Sciences Sector, Postgraduate Program Interdisciplinary in Community Development, State University of Midwest, Guarapuava 85040-167, Paraná, Brazil; julara2008@hotmail.com; 2Laboratory of Food and Nutrition, Postgraduate Program of Biotechnology, Faculty of Pharmaceutical Sciences, Food and Nutrition, Federal University of Mato Grosso do Sul, Campo Grande 79070-900, Mato Grosso do Sul, Brazil; elisvania@gmail.com

**Keywords:** food choices, healthiness, front labeling

## Abstract

This research aimed to evaluate the effect of interdisciplinary educational interventions on children’s attitudes, knowledge, preferences, and perceptions about different nutrition labels. Four hundred and ten elementary school children, aged between seven and ten years, participated in the research. The children completed questionnaires on attitudes, knowledge, and preferences about nutrition labeling and on perceived healthiness of a food product considering different types of nutrition labels (Pre-Intervention). They then participated in educational interventions as a strategy to address nutrition labeling of foods (Intervention). Finally, the Pre-Intervention questionnaires were reapplied (Post-Intervention). The intervention improved children’s attitudes and knowledge about nutrition labeling. It also showed that this public has a greater preference for labels printed on the front of the food package. The label in the form of a warning was considered the most favorable for comparing the healthiness of the food product among children, while the table was the least preferred. It is concluded that interdisciplinary educational interventions are effective in improving children’s attitudes, knowledge, preferences, and perceptions about different nutrition labels on a food product. The front label model is the most suitable for food packaging aimed at children.

## 1. Introduction

Healthy eating is associated with a better quality of life and human development. It presents more favorable effects if instituted from childhood, because children are in the learning process and are more receptive to information [1]. Nevertheless, changes in lifestyles and the increased production of processed and ultra-processed foods have altered the population’s eating patterns, caused mainly by the nutritional transition phenomenon. This period, which began in the 1950s, is marked by high consumption of foods high in energy, fat, sugar, and sodium. Additionally, it is marked by the reduced intake of fruits, vegetables, and dietary fiber, in addition to a low level of physical activity [2]. This type of eating habit is generally associated with higher prevalence of chronic non-communicable diseases (NCDs) [3], such as diabetes, hypertension, cardiovascular diseases [4], dyslipidemia [5], liver diseases [6], and obesity [7]. Studies have shown that these diseases can begin in childhood [8] and adolescence [9], continuing through subsequent stages of life.

Food products targeting children are commonly marketed using marketing strategies associated with fun, since they attract children’s interest in buying [10,11,12]. This form of advertising creates positive associations between the packaging and the food, which can influence food choices. Generally, the products have colorful packaging, collectible items, and/or cartoon characters, with emphasis on the front of the label [13]. However, studies have shown that most products intended for children have an unfavorable nutritional profile for human growth and development [11,14,15,16,17]. The food categories with the most attention to children are sweets, dairy products, fast food [17], breakfast cereals, cookies, and snacks [11]. In this context, Zucchi and Fiates [14] found that 88% of packaged foods intended for children are ultra-processed. Furthermore, 70% of the children’s commercials advertised products with high levels of sugar, saturated fat, trans fat, and/or sodium [15], well above the recommended daily intake recommended by the World Health Organization (WHO) [16].

At the end of the 20th century, there was a worldwide mobilization to implement different front-of-pack nutrition labeling systems, known as semi-directive and directive [18]. Semi-directive labels are those with visual characteristics, such as colors and/or symbols, that help communicate the healthiness of critical nutrients, such as total fat, saturated fat, trans fat, sugar, and sodium. An example of this methodology is the traffic light system, which uses the colors green, yellow, and red to inform of low, medium, and high nutritional content, respectively [19,20]. In the case of directional labels, direct and brief icons of the wholesomeness of a food product are used, without presenting detailed information, as is the case with the nutrition alert (warning) system. In this method, the expressions “High” or “Excess” in critical nutrients are presented in the forms of black symbols, such as an octagon, a triangle, and a magnifying glass [19,20,21,22]. In Brazil, frontal nutrition labeling in the nutrition alert model became mandatory only in 2020, and companies would have a period of two years to adapt to the new regulations [23,24]. However, so far, few companies have adhered to the use of this type of labeling, which hinders the development of new research in the area, especially among children.

The front nutrition labeling models make it easier for the consumer to comprehend [25] the healthiness of an alimentary product [21]. This form of presentation makes the excess of nutrients visible and informative, being able to modify the associations caused by the packing [26]. Evaluations about the influence of the front labeling with children showed positive results, improving the perception and awareness about the nutritional profile of the products [10,27,28]. These data support suggests that the frontal labeling of food products is the best form of presentation, as it helps in the evaluation and purchase intention of the product [29], which makes this model more suitable for children [30,31]. It is noteworthy that the older label in table format is reported to be difficult to identify and understand, having little efficiency, especially among individuals with low literacy level [32], such as children. Thus, it is essential to implement educational actions that promote healthier food choices from childhood, ensuring adequate [33] and persistent nutrition throughout life [34].

Some research has shown that actions involving nutrition labeling with children are beneficial for improving learning about the subject, among other effects [35,36,37]. Katz et al. [37] noted that actions that motivate and encourage children to read food labels contribute to improved nutritional literacy and food choices. In addition, children who receive training on the subject have a greater perception of the value of nutrition labeling as an important factor when purchasing food [36]. They also have greater nutritional knowledge [35,36], increase their frequency of reading the nutritional label of industrialized products, and improve their behavior about food [35]. However, more studies are needed with this audience that uses interventionist educational activities to improve understanding and knowledge about food labeling. Interdisciplinary educational interventions can be efficient in this context, since they seek comprehensive training of the children. This action is considered motivating, innovative, and enriching for learning because it includes collaborative activities, actively involving children and improving the interaction of contents from different areas [38]. In this context, the present study aimed to use interdisciplinarity as a teaching proposal, approaching different contexts of basic nutrition, dietetics, public policies, statistics, teaching pedagogy, and graphic design to evaluate the effect of educational interventions on children’s attitudes, knowledge, preferences, and perceptions about different nutritional labels.

## 2. Materials and Methods

### 2.1. Ethical Issues

The research protocol was approved after a full board review by the Research Ethics Committee of the University (number 4.471.022/2020), which indicated the use of the written Informed Consent Form (ICF). This form was signed by the parents or guardians of the children. Additionally, it was necessary for the children to sign the Free and Clarified Consent Form (FCCF), agreeing to their participation. The inclusion criteria for participation were: school-age children regularly enrolled in the 2nd, 3rd, 4th, and 5th grades in municipal schools in Guarapuava, PR, Brazil, as well as participation in all stages of the research and signing of the ICF by the guardians and the FCCF by the schoolchildren. The exclusion criteria were: age below or above the established limit, children not enrolled in the schools and grades selected, missing any of the research stages, incomplete completion of the evaluative instruments, and failure to present the signed ICF or FCCF.

### 2.2. Type of Study, Sampling, and Target Audience

This is an experimental/intervention study. The research was conducted in the year 2022 with a representative sample of the population of children (N = 8332 subjects) of school age (7–10 years), enrolled between 2nd and 5th grade in public schools in Guarapuava, PR, Brazil. The sample was determined in 2 stages: (1) the schools were selected through non-probabilistic sampling by convenience, 5 being chosen, being the largest number of students; (2) after the selection of schools, the children were chosen through simple random sampling, considering the following parameters: total number of students enrolled in the 2nd to 5th grades in city schools, confidence level of 95%, and maximum error of 5%, totaling a minimum representative sample of 239 students. A total of 649 children were invited to participate in the study. Of this total, 143 were excluded for not delivering the ICS signed by the legal guardian. The remaining 506 children agreed to participate in the survey and signed the FCCF. Ninety-six were excluded for not completing the questionnaires in full or for not participating in some of the stages of the research. Thus, a total of 410 individuals participated effectively, 71.6% more than the minimum required sample. The children’s racial data were collected directly from the registration form available at the school, which was completed by parents or guardians at the time of enrollment. The list of races available on the form were the 5 used by the Brazilian government: fair, brown, dark brown, Asian, and indigenous.

### 2.3. Study Design

The research was organized in 3 stages: Pre-Intervention, Intervention, and Post-Intervention. In the Pre-Intervention, children completed questionnaires to evaluate attitude, knowledge, and preferences about nutrition labeling and perceived healthiness of food products considering different types of nutrition labels. After 2 weeks, the Intervention began, referring to the interdisciplinary educational interventions. Topics related to nutritional labeling of foods were addressed, which included theoretical and practical activities on the topics: (a) concepts about nutritional labeling and (b) types of nutritional labels (table, traffic light, and alert—Figure 1). These activities were carried out for 2 consecutive weeks. In the Post-Intervention, the children filled out the same questionnaires applied in the Pre-Intervention, which allowed for a quantitative evaluation of learning in the period (Figure 2).

#### 2.3.1. Pre-Intervention Stage

The children answered 2 questionnaires regarding attitudes, knowledge, and preferences about nutrition labeling and about the perception healthiness of food products considering different types of nutrition labeling (Appendix A). Both instruments were presented in Brazilian Portuguese, the native language of the participants, being self-administered and answered in the classroom. The researchers assisted the children with basic instructions to answer the questionnaires, but without interfering with the answers. It is noteworthy that children in the age group of the study (7–10 years) have similar cognitive development [39]. Furthermore, the instruments were developed with an appropriate level of difficulty for this age group, aiming at the children’s understanding. The objective of these evaluations in the Pre-Intervention stage was to diagnose the understanding and difficulties of the participants in relation to the topics addressed, which helped to guide the educational action. The average time to complete the questionnaires was 20 min. The estimation of the questionnaire’s composite reliability (congeneric measures) was 0.94.

##### Attitude, Knowledge, and Preferences about Nutrition Labeling

To evaluate children’s attitudes, knowledge, and preferences about nutrition labeling, a questionnaire adapted from Zucchi and Fiates [14], Khandpur et al. [40], and Barros et al. [41] was applied. The questionnaire was organized into 2 sections: (a) attitudes and knowledge about nutrition labeling, consisting of questions (Q) Q1 through Q16; and (b) preferences about nutrition labeling, consisting of Q17 and Q18.

*(a)* 
*Attitudes and knowledge about nutrition labeling*


The instrument consisted of 16 multiple choice questions (Q), which were: (Q1) “The nutrition label should be present on all food product packaging. Do you know what it means?”; (Q2) “Do you usually look at the nutrition label on food packages?”; (Q3) “Do you find it easy to find the nutrition label on food packages?”; (Q4) “Do you find it easy to understand the nutrition label information on food packages?”; (Q5) “Can you tell if a food product is healthy or not just by looking at the nutrition label?”; (Q6) “Do you think it is important for a food product to have the nutrition label on the package?”; (Q7) “Do you know why a food product needs to have the nutrition label on the package?”; (Q8) “Do you pay attention to the nutrients on the nutrition label on food product packaging?”; (Q9) “Do you think that foods rich in sugar, fat and sodium are bad for your health?”; (Q10) “Do you find it easy to understand the list of ingredients present on food product packaging?”; In Q11, Q12, and Q13, children had to look at the nutrition label in the form of a table, traffic light, and alert, respectively, and they had to indicate “where the nutrition label should appear on food product packaging”; (Q14) “If you notice that the food product has nutrients that are bad for your health, what do you do?”; (Q15) “Do you think it is important to look at the ingredients list of a food product (what it is made of)?”; (Q16) “If you checked yes in the previous question (15), why do you think it is important to look at the ingredients list of a food product?”.

To answer Q1 through Q10 and Q15, the children were to choose: “yes”, “no”, or “maybe/sometimes”. For Q11 to Q13, one of the following options should be selected: “Behind”, “Above”, “On the side”, “In front”, “Below”, or “Don’t know”. For Q14, the answers were: “Eats anyway”, “Doesn’t eat”, and “Don’t know”; while, for Q16, the participants had to select one of the following alternatives: “To see if it is healthy”, “To see if it is bad for your health”, or “Other–describe”. The illustrative images from Q11 to Q13 were designed using the tools Paint and GNU Image Manipulation Program (GIMP) (version 2.10.32), so as not to correspond to any commercial product known to the children, avoiding possible interference in the evaluation.

*(b)* 
*Preferences on nutrition labeling*


This section consisted of two questions (Q), namely: (Q17) “Do you think it is cooler for the nutrition label to appear in which place on the package?” and (Q18) “Which of these three nutrition labels do you think is coolest to appear on the package?”. To answer Q17, one of the following options had to be selected: “Behind”, “Above”, “On the side”, “In front”, “Below”, or “Don’t know”; for the last question (Q18), the answers were: “Table”, “Traffic light”, or “Alert”. The illustrative images for Q17 and Q18 were designed using the tools Paint and GIMP, so as not to correspond to any commercial products known to the children, avoiding possible interference in the evaluation.

The evaluation of the nutrition labeling attitudes, knowledge (“section a”, Q1 to Q16) and preferences questionnaire (“section b”, Q17 and Q18) occurred as follows: Questions Q1 to Q10 and Q14 to Q18 were evaluated in relation to the frequency (%) of marking for each alternative. In order to better classify the interpretation and evaluation of the effect of the interdisciplinary educational intervention, the alternatives “no” or “maybe/sometimes” were combined and considered inadequate answers about knowledge of nutritional labeling. In Q11, we point out that the Brazilian legislation [24] does not stipulate an exact place for the insertion of the nutrition label in the table model. Thus, the correct answer was considered when the child marked all the options (behind, above, on the side, in front, below), being evaluated by frequency (%). Regarding Q12 and Q13, the frequency (%) of identification for the correct answer “in front” was evaluated.

##### Perception of Healthiness among Different Nutrition Labels

The questionnaire was adapted from Poquet et al. [31], Lima et al. [27], and Reynolds et al. [42]. In the questionnaire, a commercial product (filled sweet cookie) was presented in 2 versions (A and B), and version A was considered healthier than version B. However, both types of the product were high in at least one critical nutrient (total fat, saturated fat, trans fat, sugar, and/or sodium). The flavor and serving size characteristics did not vary across product presentations. For each version, 3 types of labels were presented (table, traffic light, and alert).

The 3 types of nutrition labels presented in the questionnaire were prepared by adapting the packaging available on commercial products, containing fictitious nutrient values. The label in the nutrition table format was composed of the declaration of the amounts and percentages of daily values (%DV) for energy, carbohydrate, protein, total fat, saturated fat, trans fat, dietary fiber, and sodium. In addition, it included serving size information, home measurement, and the list of ingredients [23,24]. For the traffic light label, quantitative food information related to nutritional content per serving was included, referring to calories, sugar, total fat, saturated fat, and sodium. There were also text descriptors (low, medium, and high) and color coding (green, yellow, and red) that indicated the nutritional content of the product. This classification considered the criteria established by the Food Standards Agency [43]. The label in the nutrition alert format used a black rectangular symbol with a magnifying glass, which identified the high content of saturated fat, sodium, and/or added sugar, nutrients considered critical because they are harmful to health. The symbol was included when the content of added sugar (≥15 g/100 g of food), saturated fat (≥6 g/100 g of food), and sodium (≥600 mg/100 g of food) exceeded the criteria established by Brazilian legislation [24].

To evaluate the children’s perception of the healthiness of the products, a 5-point Likert scale was used, as follows: “1—Very unhealthy”, “2—Unhealthy”, “3—Neither healthy nor unhealthy”, “4—Healthy”, and “5—Very healthy”. The participants were instructed to mark the alternative that best represented the information on the illustrated nutritional label. The results were expressed through mean scores, with scores 1, 2, and 3 being considered perceptions that were unhealthy, while scores 4 and 5 were classified as healthy.

#### 2.3.2. Intervention Stage—Educational Interventions

In this stage, two educational activities were carried out over a period of 2 consecutive weeks. Each action lasted an average of 30 min. In the first intervention, basic concepts of food nutrition labeling were presented. In the second, the children were explained the types of nutrition labels (table, traffic light, and alert). The themes addressed in the educational interventions are described in Table 1, and they were organized by means of the literature adaptations [29,37,44,45,46,47]. Brazilian laws [23,24] and other governmental documents were also used [48,49]. The theoretical and practical educational interventions totaled 30 min each, 10 min of dialogic exposure and 20 min of games or play. For the practical part, playful dynamics (games and jokes) and easy-to-understand games were used to facilitate the teaching–learning process.

#### 2.3.3. Post-Intervention Stage

The questionnaires applied in the Pre-Intervention were reapplied 15 days after the last training to assess the effect of learning (Intervention stage) on the children’s understanding and knowledge of the nutrition labeling topic. Aiming to expand the children’s learning in the long term, material with basic information on how to interpret and understand the nutritional labeling of foods was also handed out.

### 2.4. Statistical Analysis

The results were evaluated using frequency, mean, and standard deviation. The dependent Student’s *t*-test was used to compare the means of children’s responses to in the questionnaire of healthiness perception among different nutrition labels, between Pre- and Post-Intervention stages. In this same questionnaire, the comparison of means between the different versions of the same label (A and B) was performed using the independent Student’s t test. In addition, Tukey’s test was applied to compare the means between the three nutritional label versions. The McNemar test was used to evaluate the effect of educational interventions on children’s attitudes, knowledge and preferences about nutrition labelling, between Pre- and Post-Intervention stages. All results were analyzed using the R software, version 4.0.4, considering a 5% significance level (*p* < 0.05).

## 3. Results

The average age of the 410 participating children was 8.79 years (±0.89). The sample evaluated was 47% (192) male and 53% (218) female. Regarding schooling, 4% (15) were in 2nd grade, 37% (153) in 3rd grade, 20% (81) in 4th grade, and 39% (161) in 5th grade. Regarding race, most children were classified as brown (49%, *n* = 201), followed by fair (41%, *n* = 168) and dark brown (10%, *n* = 41).

Overall, the educational intervention improved attitudes and knowledge about nutrition labeling (*p* < 0.05), except for Q8 and Q10 (*p* > 0.05). It is worth noting that children think it is important that the nutrition label is presented on the package (*p* < 0.05) and that the ingredients list be read (*p* < 0.05), as can be observed in Q6 and Q15, respectively (Table 2 and Table 3). Thus, it demonstrates the relevance of continuous and early interventions since childhood, so that learning can last in the subsequent phases of life. Although the educational interventions showed a positive effect for Q11 (*p* < 0.05), the result was not so expressive in increasing the children’s knowledge about the location of the nutrition label in table form (Table 2).

The intervention increased the number of children who would not eat a food product if they knew it was bad for their health (*p* < 0.05) (Figure 3). Table 3 shows the frequency of participants’ responses to Q15 and Q16 on attitudes and knowledge about nutrition labeling.

The educational intervention provided a greater number of children who understood it was important to look at the list of ingredients on products (Q15). In addition, there was an increase in children reporting a need to look at the list to see if the food is healthy and/or if it is bad for their health. The results of Q17 and Q18, related to nutrition labeling preferences, are shown in Figure 4.

Children’s preference for the location of the nutrition label on the package (Q17) was significantly influenced by the educational intervention. Thus, there was an increase in the number of children who preferred the label appearing on the front of the package. For the other sites, there was a reduction in choice by children (*p* < 0.05). Regarding the type of labeling (Q18), the Intervention increased the number of participants with a preference for the traffic light, reducing the choice for the table (*p* < 0.05). There was no change in the preference for the alert label (*p* > 0.05). Table 4 shows the average scores for the healthiness perception questionnaire among the different nutrition labels.

The mean scores assigned by the children for all product labels were 1 (not healthy) and 2.5 (unhealthy) at both the Pre- and Post-Intervention. This shows that this public already had an understanding that the sweet stuffed cookie is not a healthy food, considering that both versions of the labels used in the research could not be classified as healthy. The perceived healthiness between label versions in table form in the Pre-Intervention showed no significant difference (*p* > 0.05). However, higher scores were assigned (*p* < 0.05) for version A of the traffic light and alert labels. In the Post-Intervention, only the alert system showed a significant difference, with a higher score for version A. Overall, the educational intervention improved perceived healthiness by reducing children’s scores for the version B of the table and alert labels and the version A of the traffic light label. The average scores for comparing the versions between the different types of labels are shown in Table 5.

There was no significant difference (*p* > 0.05) between the types of nutrition labels for version A of the sweet, filled cookie in the Pre-Intervention. However, the mean healthiness scores for version B of the labels were lower (*p* < 0.05) for the alert system, followed by the traffic light system, at both stages. At Post-Intervention, version A of the product had statistically lower scores for the traffic light system, and this did not differ between the table and the alert.

## 4. Discussion

The present study demonstrated that educational interventions improve attitudes and knowledge about nutrition labeling, corroborating similar studies [37,50]. When teaching employs theoretical didactics in pair with playfulness, it contributes to the biopsychosocial development of students [51]. Moreover, the use of playful activities, such as games and jokes, contribute to improve the teaching-learning process in childhood [52]. This strategy uses simple instruments that arouse the children’s interest in the proposed theme. They can also be used to associate disciplinary content in a more concrete and attractive way [53], stimulating the search for knowledge, reflection, and collaborative actions [52]. In this way, it is possible to broaden child development by relating different cognitive, affective, social, linguistic, motor [53], and sensory areas [54]. These lived experiences, together with the interaction with the environment, promote different stimuli. Then, they are converted into electrical/chemical impulses sent to the thalamus, a structure in the central region of the brain, responsible for processing the information [54]. This process promotes dynamic learning, which gives pleasure in the act of learning [53].

Despite studies showing the effectiveness of educational actions, this effect was not observed for Q8 and Q10, whose themes were related to the observation of nutrients and the list of ingredients on the package. It is possible that other factors interfere in the purchase of foods, especially when the packaging is attractive, which can prioritize the child’s attention [11,55] on elements other than those related to healthiness. Moreover, sensory characteristics can influence food consumption, such as product taste, aroma, and appearance [56,57]. Added to this is the use of technical terms in ingredient lists, which are reported to be difficult for adult consumers to understand [58]. Consequently, the interpretation process may be even more complex for children, as neurodevelopmental processes are growing and evolving [39].

The educational intervention had a positive effect on children’s knowledge about the location of nutrition labels on packages. However, for the label in table form, this result was not so expressive. According to Anitha and Anusuya [59], schoolchildren find this model of labeling complicated and time-consuming to interpret. Thus, for more complex questions, it is possible that a greater number of interventions or the use of strategies with more specific approaches may promote better results, as demonstrated by Katz et al. [37]. The authors observed that encouraging children to look for and interpret labels on actual food products improves their understanding of nutritional information in a table format. Furthermore, they concluded that involving the school and the family in this educational process also brings positive effects.

The increase in the number of children who would not eat a food product if they knew it was harmful to health may be related to the fact that they had low understanding of which nutrients/foods could be harmful to health. A similar effect was found by Olivares et al. [60], who found that elementary school children would not buy food products that had the alert system on the packaging indicating excess of critical nutrients. Despite this, children may not apply this knowledge in their eating habits, since they believe they do not need to worry about eating, besides being influenced by the taste of food [56]. Thus, it is important that nutritional educational interventions are not only directed to nutritional knowledge, but also involve changing children’s attitudes and eating habits [56,61,62].

The participants’ responses to Q15 and Q16 demonstrate the importance of the presence of the ingredients list in food products. This information, along with the nutritional label, can facilitate food choices, contributing to improved health perception and awareness of the nutritional profile of food products, as found in other studies [10,27,37].

After the educational intervention, there was an increase in the number of children who preferred the label appearing on the front of the package. Other authors have also shown that front labeling is considered by children and adults to be easy to use, more interpretive and relevant, visually attractive [63] and simplified, allowing for quick comparisons between food products [64]. Furthermore, Olivares et al. [60] observed that more than 70% of the evaluated children liked to be informed about the nutritional content of foods by the alert method. Regarding the type of labeling, most participants preferred the traffic light. This option may be related to the presence of colors and ease of interpretation and use of this type of labeling. The colors are considered aesthetically pleasing, standing out from other nutrition label designs [63]. In addition, the color coding used (green, yellow, and red) is easily interpreted, since this system is already commonly used for vehicle and pedestrian traffic control [64].

The results of this study show that the label in table form is the least suitable for children, since the children could not differentiate which version was more or less healthy. This effect may be related to the complexity of the numeric form of presentation of data for this type of label, which makes it difficult to interpret the information in general, causing some confusion [19]. Traffic light and alert labels, on the other hand, are types of labels that have greater acceptance by children [28,63], which was also found in this study. Despite this, some authors have concluded that children and adults may have difficulty determining the wholesomeness of a product in the traffic light system, especially when the product does not have most of the indicators in the colors red or green [64]. In contrast, the alert model appears to be the most favorable for comparing wholesomeness between similar products [10,27,28,65]. This effect corroborates the findings of the present research, as participants were able to identify the least healthy product (version B) in the alert system, both in the Pre- and Post-Intervention stages. According to Hock et al. [65], the alert system is easily understood due to the use of a simple symbol that is recognizable by most of the population. Despite the positive results obtained in the present research, it is clear that only two educational activities for school children are not enough to fully improve the perception of healthiness for the different types of nutrition labels used.

The data shown in this study reinforce that children have an easier time interpreting nutrition labels when the information is explicit in the form of an alert and a traffic light, being superior for the first type of label. In any case, frontal labeling is the most suitable for use and interpretation in children’s food choices, since they are easy to identify and visually attractive [63]. They also contribute to reducing the positive health association with products that are high in fat, sodium, and sugar [28].

## 5. Conclusions

Interdisciplinary educational interventions are effective in improving children’s attitudes, knowledge, preferences, and perceptions about different nutrition labels on a food product. Among the types of labeling, the frontal model, especially the alert, is the most accepted among children, presenting a greater perception of healthiness.

As strengths, this study allowed us to verify that interdisciplinary interventions, even if carried out in a short period, can effectively contribute to children’s learning about nutritional labeling. Thus, it is believed that strategies carried out over a long period and with greater depth may bring even more promising contributions to learning and child development on the subject of healthy eating. In addition, there are few studies that address the theme of nutrition labeling for children. In this way, this study can help and stimulate other researchers in the use of new interdisciplinary strategies that can broaden the understanding of nutritional labeling, improving the food choices of this public.

The COVID-19 pandemic is a limiting factor in this research, since longer periods for carrying out educational activities were not allowed by schools. There was also a certain restriction on children since it was necessary to maintain social distance, which makes more individual explanations difficult. The evaluation of only one food product can be considered another limiting factor of the study. In this aspect, it is possible that the children already had some prior knowledge that cookies were not so healthy before carrying out the educational interventions. This fact may have contributed to the lower scores in the children’s evaluation of the healthiness of the product in the Pre-Intervention stage.

New studies are suggested that may consider the application of interdisciplinary educational actions in COVID-19-free environments and that involve a longer period of time, in addition to family members and teachers. Additionally, it is important to study the healthiness of a wider variety of food products, including healthy ones. This could expand evaluations with children, proposing new public policies aimed at healthier food consumption throughout life.

## Figures and Tables

**Figure 1 foods-12-04290-f001:**
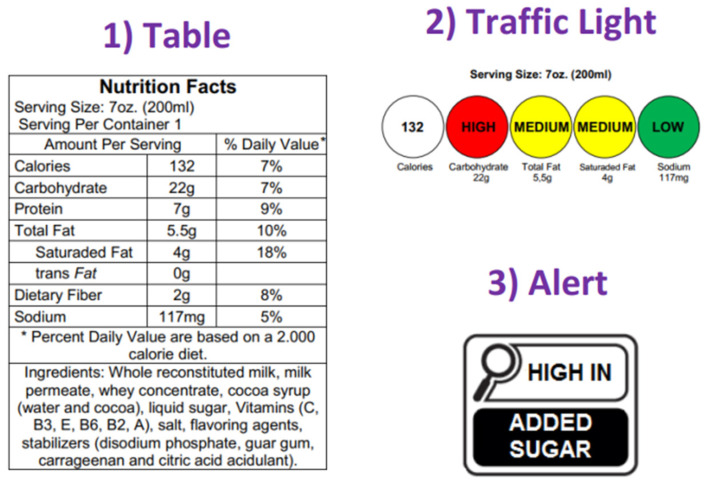
Examples of nutritional labels (table, traffic light, and alert) used in questionnaires and interdisciplinary educational interventions.

**Figure 2 foods-12-04290-f002:**
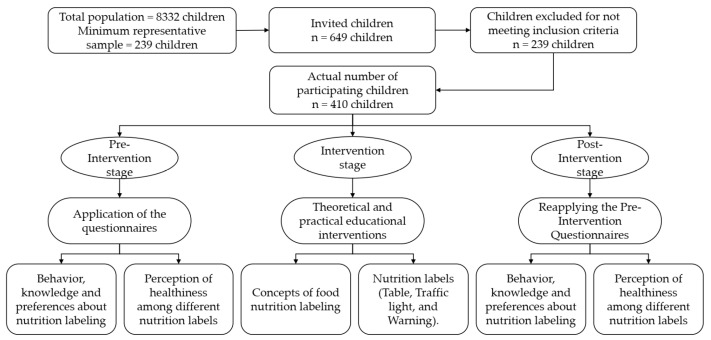
Flowchart of the study.

**Figure 3 foods-12-04290-f003:**
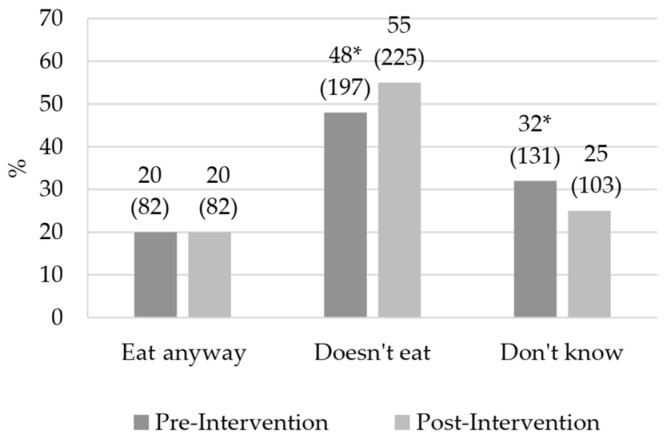
Effect of educational interventions on children’s attitudes and knowledge about nutrition labeling for question 14 (Q14). *n* = 410; * Values indicate significant difference by McNemar’s test (value of *p* ≤ 0.05); Question 14 (Q14) “If you notice that the food product has nutrients that are bad for your health, what do you do?”.

**Figure 4 foods-12-04290-f004:**
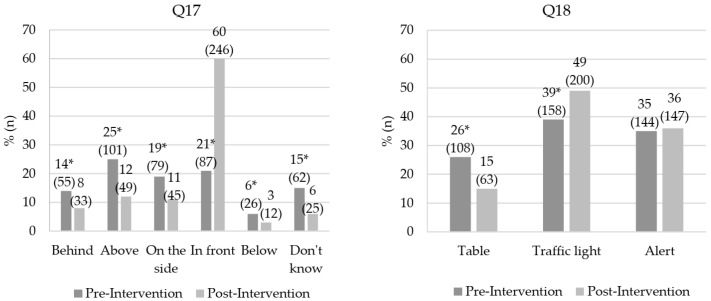
Effect of educational interventions on children’s nutrition labeling preferences for questions 17 (Q17) and 18 (Q18). *n* = 410; * Values indicate significant difference by McNemar’s test (value of *p* ≤ 0.05); Question (Q); Q17. “Do you think it is cooler for the nutrition label to appear in which place on the pack-age?”; Q18. “Which of these 3 nutrition labels do you think is coolest to appear on the pack-age?”.

**Table 1 foods-12-04290-t001:** Themes covered in the theoretical–practical educational interventions.

Week	Activity	
	Theoretical	Practice
1	Concepts and importance of nutrition information and function of the ingredients list, information present on nutrition labels (energy value, carbohydrate, protein, fat, sodium, etc.), and the relationship between nutrition information and non-transmissible chronic diseases [23,44,48].	The “nutrition labeling trail” game: The trail was made up of distant circular areas with different colors. Each color was responsible for a function: red—questions to test knowledge; green and yellow—deciphering a problem; and black—luck or setback. The topics covered were about nutritional labeling. The child had to roll a number die, which directed how many areas they had to walk. In the selected area, and it was necessary to perform the corresponding activity. If the child completed it successfully, they remained on site, otherwise they returned two areas. The objective of the game was to cross the finish line in a fun way, as well as to facilitate learning [46,47].
2	Review of concepts covered in the first intervention, types and location of nutrition labels (table, traffic light, and alert), and interpretation of nutrition information for choosing healthier foods [23,24,44,49].	“Packaging ranking” game: fictitious packages of industrialized products common to children (filled cookies, chocolate drinks, and snacks) were used. Each product was presented in three similar packages, but with different nutritional contents. For the stuffed cookie and the snack, the labeling consisted of the nutrition table and traffic light. For the chocolate drink, the nutritional information consisted of the table and the alert. The children were instructed to analyze each product for nutritional content, considering the three packages individually. Then, they should rank the products from healthiest to least healthy [29,37,45].

**Table 2 foods-12-04290-t002:** Effect of educational interventions on children’s attitudes and knowledge about nutrition labelling for questions 1 (Q1) to 13 (Q13).

Question	Pre-Intervention				Post-Intervention				*p* *
	Yes		No/Maybe/Sometimes		Yes		No/Maybe/Sometimes		
	*n*	%	*n*	%	*n*	%	*n*	%	
Q1	133	32	277	68	261	64	149	36	0.000
Q2	120	29	290	71	162	40	248	60	0.001
Q3	209	51	201	49	249	61	161	39	0.002
Q4	119	29	291	71	145	35	265	65	0.045
Q5	189	46	221	54	226	55	184	45	0.006
Q6	346	84	64	16	376	92	34	8	0.000
Q7	151	37	259	63	302	74	108	26	0.000
Q8	173	42	237	58	190	46	220	54	0.213
Q9	297	72	113	28	360	88	50	12	0.000
Q10	226	55	184	45	244	60	166	40	0.220
	**Correct**		**Incorrect/Not Sure**		**Correct**		**Incorrect/Not Sure**		***p* ***
	** *n* **	**%**	** *n* **	**%**	** *n* **	**%**	** *n* **	**%**	
Q11	8	2	402	98	46	11	364	89	0.000
Q12	25	6	385	94	374	91	36	9	0.000
Q13	23	6	387	94	370	90	40	10	0.000

*n* = 410; Question (Q): Q1. “The nutrition label should be present on all food product packaging. Do you know what it means?”; Q2. “Do you usually look at the nutrition label on food packages?”; Q3. “Do you find it easy to find the nutrition label on food packages?”; Q4. “Do you find it easy to understand the nutrition label information on food packages?”; Q5. “Can you tell if a food product is healthy or not just by looking at the nutrition label?”; Q6. “Do you think it is important for a food product to have the nutrition label on the package?”; Q7. “Do you know why a food product needs to have the nutrition label on the package?”; Q8. “Do you pay attention to the nutrients on the nutrition label on food product packaging?”; Q9. “Do you think that foods rich in sugar, fat and sodium are bad for your health?”; Q10. “Do you find it easy to understand the list of ingredients present on food product packaging?”; Q11. “Look at the nutrition label below (Table) and indicate where the nutrition label should appear on food product packaging”; Q12. “Look at the nutrition label below (Traffic Light) and indicate where the nutrition label should appear on food product packaging”; Q13. “Notice the nutrition label below (Alert) and indicate where the nutrition label should appear on food product packaging”; * McNemar’s test, considering significant the *p* value ≤ 0.05.

**Table 3 foods-12-04290-t003:** Effects of educational interventions on children’s attitudes and knowledge about nutrition labeling for questions 15 (Q15) and 16 (Q16).

Question/Answer	Pre-Intervention		Post-Intervention		*p* ***
	*n*	%	*n*	%	
Q15					
Yes	225	55	295	72	0.000
No/Maybe/Sometimes	185	45	115	28	
Q16 *					
To see if it is healthy	126	31	159	39	0.014
To see if it is bad for your health	83	20	125	30	0.000
Other **	16	4	11	3	0.359
Did not answer Q15	185	45	115	28	0.000

*n* = 410; Question (Q): Q15. “Do you think it is important to look at the ingredients list of a food product (what it is made of)?”; Q16: “If you checked yes in the previous question (Q15), why do you think it is important to look at the ingredients list of a food product?”; * Obs. Only children who marked “yes” in question 15 (Q15) answered this question; *n* = 225 (Pre-Intervention), *n* = 295 (Post-Intervention). ** In this option, most answers were “curiosity”, “to see what the product is made of” and “to see if it gives allergies”, among others; *** McNemar’s test, considering significant the value of *p* ≤ 0.05.

**Table 4 foods-12-04290-t004:** Mean scores (± standard deviation) obtained in the questionnaire of “healthiness perception among different nutrition labels”, considering the Pre- and Post-Intervention stages.

Label	Pre-Intervention	Post-Intervention
Table		
Version A	2.5 ± 1.10 ^aA^	2.4 ± 1.00 ^aA^
Version B	2.5 ± 1.07 ^aA^	2.3 ± 1.05 ^bA^
Traffic light		
Version A	2.5 ± 1.30 ^aA^	2.0 ± 1.04 ^bA^
Version B	2.1 ± 1.15 ^aB^	1.9 ± 0.98 ^aA^
Alert		
Version A	2.3 ± 1.19 ^aA^	2.2 ± 0.94 ^aA^
Version B	1.7 ± 1.07 ^aB^	1.3 ± 0.61 ^bB^

*n* = 410; Distinct lower case letters in the same row indicate significant difference between the Pre- and Post-Intervention stages analyzed by Student’s dependent *t*-test (*p* < 0.05); Distinguished capital letters in the same column indicate significant difference between the different versions of the same label (A and B) analyzed by independent Student’s *t*-test (*p* < 0.05); Version A of the product was considered healthier than version B; Scale of scores: 1. Not healthy; 2. Unhealthy; 3. Neither healthy nor unhealthy; 4. Healthy; and 5. Very healthy.

**Table 5 foods-12-04290-t005:** Mean scores (±standard deviation) for the different label versions obtained in the “perceived healthiness among different nutrition labels” questionnaire.

Label Versions	Table	Traffic Light	Alert
Pre-Intervention			
Version A	2.5 ± 1.10 ^a^	2.5 ± 1.30 ^a^	2.3 ± 1.19 ^a^
Version B	2.5 ± 1.07 ^a^	2.1 ± 1.15 ^b^	1.7 ± 1.07 ^c^
Post-Intervention			
Version A	2.4 ± 1.00 ^a^	2.0 ± 1.04 ^b^	2.2 ± 0.94 ^a^
Version B	2.3 ± 1.05 ^a^	1.9 ± 0.98 ^b^	1.3 ± 0.61 ^c^

*n* = 410; Distinguished letters in the same row indicate significant difference by Tukey’s test (*p* < 0.05); Version A of the product was considered healthier than version B; Scale of scores: 1. Not healthy; 2. Unhealthy; 3. Neither healthy nor unhealthy; 4. Healthy; and 5. Very healthy.

## Data Availability

The data used to support the findings of this study can be made available by the corresponding author upon request.

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
