# Peer review of "How Interdisciplinary Interventions Can Improve the Educational Process of Children Regarding the Nutritional Labeling of Foods"

_foods, 2023, doi:10.3390/foods12234290_

Round 1
Reviewer 1 Report
This manuscript is in strong shape and will surely be of interest to Foods readers. Although I do have some comments/suggestions, they're fairly minor and mainly focused on clarifying study details.
For the sake of comprehensibility, it would be helpful to add visualizations of a table, traffic light, and alert system. These systems/symbols aren't common in every country, and I personally found myself Googling to make sure I understood what the authors meant with the use of each term. It could specifically be helpful to include examples of the labels used in the study questionnaire to help readers fully gauge the nature and, where relevant, difficulty of the tasks.
Additionally, if any room remains for further reflection after incorporating my notes and those of the other reviewers, I'd be interested in the authors dedicating more space in the Discussion to considering how remarkable it is that the results looked so promising after only an hour of intervention. Imagine how much learning and development could take place (or long-term retention) with a more in-depth intervention.
Secondary notes:
- What does 8.332 stand for on line 107? Can you clarify?
- Did the two intervention sessions take place over "2 consecutive weeks" as described on line 132 or "a period of one month" as described on line 227? Edit for clarity and to help with potential replicability.
- There's some room to clarify the questionnaire scoring.
-- For question 16, could children select more than 1 response option?
-- The "Evaluation of nutrition labeling" section implies that the questionnaires were scored via frequencies in a neutral fashion where no answer was better than any other, but clearly the authors later do interpret "yes" responses as good/correct and "no/maybe" responses as worse/incorrect--so attributing more value to child responses than initially implied. Clarify how earlier how the scores will be interpreted and provide a rationale for combining no/maybe. I agree with what I assume is the authors' rationale for their scoring decisions, but it'll be easier for readers to follow along if this section is edited for greater transparency/clarity.
-- If students were selecting 1's and 2's when rating the cookie's healthfulness, how were their mean scores ever get above 2.0? How was this activity scored?
- Is there a reason why the grade distribution of participants was so uneven? If yes, can you elaborate--even if only as a footnote?
- For the nutrition label rating task, is it possible that you suffered from floor effects because, even prior to the intervention, children knew that cookies are typically unhealthy? Should future researchers aim to replicate this task using other foods - perhaps a wider variety of foods with some examples of foods that can vary dramatically in healthfulness like popcorn. Food for thought as a potential fruitful future direction (I see the puns).
Reviewer 2 Report
Interesting manuscript entitled "How interdisciplinary interventions can improve the educational process of children on nutrition labeling of foods"
I think it is necessary to correct/strengthen some aspects to improve the document:
1. Explain what "interdisciplinary educational interventions" means and how it differs from traditional educational interventions
2. Remove the references "Tables" and "Figures" in the discussion
3. Add the strengths and weaknesses of the study
4. In Chile, which incorporated frontal labeling in 2016, there will not be any interesting information in groups of schoolchildren/adolescents regarding education in frontal labeling, which could be useful in the discussion.
Round 2
Reviewer 2 Report
The authors of the manuscript made the requested changes. I agree with the new version
Author Response
The reviewer verified the manuscript with the requested changes and agreed with the new version.